

# Characterization of bioactive and fruit quality compounds of promising mango genotypes grown in Himalayan plain region

Neetu Saroj[1], K. Prasad[2], Sanjay Kumar Singh[3], Vishal Kumar[4], Shubham Maurya[1], Poonam Maurya[1], Rahul Kumar Tiwari[5], Milan Kumar Lal[5] and Ravinder Kumar[5]

[1] Department of Horticulture, Post-Graduate College of Agriculture (PGCA), RPCAU, Pusa, Bihar, India
[2] Department of Horticulture, Tirhut College of Agriculture (TCA), Dr. Rajendra Prasad Central Agricultural University (RPCAU), Pusa, Bihar, India
[3] Department of Plant Pathology, PGCA, RPCAU, Pusa, Bihar, India
[4] Department of Processing and Food Engineering, College of Agricultural Engineering, RPCAU, Pusa, Bihar, India
[5] ICAR, ICAR-Central Potato Research Institute, Shimla, Himachal Pradesh, India

Corresponding authors
K. Prasad, kprasad.tca@rpcau.ac.in
Ravinder Kumar, chauhanravinder97@gmail.com

## ABSTRACT

Twenty mango genotypes grown in the plains of the Himalayas were characterized by their physical, physiological, biochemical, mineral and organoleptic attributes: fruit firmness, weight, peel thickness, shape, dry seed weight, respiration rate, weight loss, and shelf life. Biochemical attributes such as soluble solids, total carotenoids, total phenolic content, antioxidant activity, titratable acidity, ascorbic acid and total sugars were also determined. In addition, mineral content and fruit-softening enzymes were measured, and an organoleptic evaluation was performed. Polygalactouronase (PG), pectin methylesterase (PME) and lipoxygenase (LOX) were measured from the pulp adjacent to the peel. Similarly, biochemical attributes and mineral content were evaluated using fruit pulp, while organoleptic evaluation included fruit pulp characters and the fruit's external appearance. The results of the study showed that the 'Malda' genotype exhibited the highest total phenolic content (560.60 μg/100 g), total antioxidant (5.79 μmol TE/g), and titratable acidity (0.37%) among the tested genotypes. 'Amrapali' had the highest soluble solid content (25.20 °B), 'Jawahar' had the highest ascorbic acid content (44.20 mg/100 g pulp), 'Mallika' had the highest total flavonoid content (700.00 μg/g) and 'Amrapali' had the highest total carotenoid content (9.10 mg/100 g). Moreover, the genotypes 'Malda', 'Safed Malda'and 'Suvarnarekha' had a shelf life of 4–5 days longer than other tested genotypes. The genotypes with high biochemical attributes have practical utility for researchers for quality improvement programmes and processing industries as functional ingredients in industrial products. This study provides valuable information on the nutritional and functional properties of different mango genotypes, which can aid in developing improved varieties with enhanced health benefits and greater practical utility for processing industries.

# INTRODUCTION

The mango is a trendy tropical fruit known for its delightful flavour, sweet taste, and vibrant colour. It is also highly regarded for its nutritional value, as it contains an array of minerals, vitamins, sugars, and fibre, along with various phytochemicals, such as polyphenols, which provide numerous health benefits (*Dars et al., 2019*; *Hu et al., 2021*; *Parvin et al., 2023*). Mango has been cultivated for over 4,000 years and all varieties are traced back to India and Southeast Asia. In India, all cultivable mangoes belong to *Mangifera indica* L., although other species such as *M. odorata*, *M. foetida*, and *M. caesia* can also be found (*Aung, 2019*). Polyphenols are organic micronutrients found in plants that are known to have unique health benefits. Mango is an excellent source of polyphenols, including mangiferin, gallic acid, gallotannins, quercetin, isoquercetin, ellagic acid, and β-glucogallin, with gallic acid being the most prevalent in the mango mesocarp (*Singh et al., 2022*). Mango is consumed in both fresh and processed forms, and both forms provide numerous health benefits. Mango has many uses, from promoting digestion to improving skin health (*Prasad, Jacob & Siddiqui, 2018*; *Singh et al., 2022*). Mango is a highly nutritious fruit with a substantial market globally.

Fruit quality attributes, such as colour, aroma, flavour, taste, and texture are of commercial importance. However, due to growing consumer interest in foods that promote health, consumer preference is now also influenced by its nutraceutical value (*Prasad et al., 2022a*). Moreover, the ripening stage during harvest and after storage has an impact on mango quality and its bioactives (*Gentile et al., 2019*). The presence of bioactive compounds in fruits is an essential indicator of fruit quality and consumption patterns. The primary polyphenols in mango are catechins, mangiferin, kaempferol, quercetin, rhamnetin, anthocyanins, ellagic acids and gallic, propyl and methyl gallate, benzoic acid, and protocatechuic acid. These compounds offer numerous preventative health benefits (antioxidative, anticarcinogenic, anti-atherosclerotic, antimutagenic, anti-inflammatory, analgesic, antidiabetic and immunomodulator) and can protect against cardiovascular diseases (*Masibo & He, 2008*; *Hu et al., 2021*). Mangiferin is one of the major phenolic components in mango pulp (around 4.4 mg/kg), seed kernel (42 mg/kg), and mango peel (1,690 mg/kg). Stem bark (71.4 g/kg). Mangiferin displays a wide range of pharmacological effects (antioxidant, anticancer, antimicrobial, anti-atherosclerotic, antiallergenic, anti-inflammatory, analgesic, and immunomodulatory) (*Masibo & He, 2008*). Mango contains a combination of polyphenols and xanthones that act as antioxidants, protecting against various ailments (*Berardini, Carle & Schieber, 2004*; *Kuria, Matofari & Nduko, 2021*), and it has a higher level of carotenoids, particularly β-carotene, than other fruits. These compounds broadly define the nutritive value of mango fruit, and it is possible to increase their levels through proper post-harvest treatments (*Hu et al., 2021*). The bioactive compounds in mango make it an excellent fruit for promoting overall health and preventing various diseases when included in a healthy diet.

Textural changes and senescence are one of the major quality concerns of mango, which leads to poor quality, and limits the shelf life (*Prasad et al., 2022a*). Deterioration of cell wall structure is associated with changes in the fruit's texture. Fruit softening is closely related to the higher activities of pectin-degrading enzyme such as PG, PME and LOX. PME hydrolyses pectin, and PG degrades galacturonic acid, and thus leading to the depolymerisation and dissolution of pectin polysaccharides (*Khaliq et al., 2017*; *Prasad et al., 2022a*). Mango fruits grown in the Himalayas' plain region are believed to be particularly rich in phytochemical contents and compounds that promote good health (*Prasad et al., 2020*; *Saroj, 2022*). However, genotype-specific profiling of bioactive compounds in these mangoes has yet to be explored extensively. To fill this knowledge gap, the present study aims to characterise the bioactive compounds in selected commercial mango genotypes grown in the Himalayan plain region. Through this investigation, we hope to gain a deeper understanding of the health-promoting properties of these mangoes and their potential as a source of bioactive compounds.

## MATERIALS AND METHODS

### Sampling and estimation

Twenty commercial mango genotypes *viz*., 'Alphonso', 'Amrapali', 'Bombai', 'Chausa', 'Dashehari', 'Gulab Khas', 'Himsagar', 'Jawahar', 'Krishna Bhog', 'Langra', 'Mahmood Bahar', 'Malda', 'Mallika', 'Prabha Sankar', 'Ratna', 'Safed Malda', 'Suvarnarekha', 'Totapuri', 'Zardalu', and 'Fazli' were obtained (Pusa, Dholi and adjoining area), RPCAU (Dr. Rajendra Prasad Central Agricultural University), Bihar. The fruits were harvested under dry weather from May–August. After harvesting, fruits were de-sapped and precooled immediately with hydro cooling to bring the temperature to a uniform level for all varieties. Fruits were stored for ripening under ambient storage conditions ($25 \pm 4$ °C and $65 \pm 5\%$ RH) for 15 days. However, the shelf life of mango genotypes (all genotypes) was completed at and within 12th of storage. The genotypes were investigated for physical, physiological, biochemical, mineral contents, organoleptic evaluation, and fruit-softening enzymes. The parameters, irrespective of their sections, were determined at the peak ripening stage. The experiment was conducted over two years, specifically in 2020 and 2022. The data presented in the study represents the average value obtained from both years.

### Determination of physical attributes

An electronic balance recorded the fruit and dry seed weight in grams (g). The thickness of the fruit peel was recorded in mm using a vernier calliper. Mango fruits' morphology (shape) was determined using International Plant Genetic Resources Institute (IPGRI) mango descriptors. A texture analyser (TA-XT Plus) was used to determine the fruit firmness, expressed as 'N' (Newton).

## Estimation of biochemical attributes

### Soluble solid contents (SSC) and total sugars

SSC Using a hand refractometer (0–50 °B) was estimated and depicted as Brix° under ambient storage. Lane and Eynon's method, described by *Ranganna (1986)*, was used to determine the total sugars.

### Titrable acidity (TA)

The procedure followed by *Singh et al. (2022)* determined the TA of mango genotypes. TA was determined by titrating against 0.1 N NaOH using a few drops of phenolphthalein indicator, which became pink and was depicted as equivalent of citric acid.

### Total carotenoid contents (TCC)

TCC was determined per the procedure followed by *Prasad et al. (2020)*. In 30 ml acetone, 10 g of pulp was homogenised until the pigment was removed entirely. The golden pigment was obtained by filling a homogenised solution in a separating funnel and washing it with petroleum ether and a pinch of sodium sulfate. For pigment separation shaken funnel was left without any disturbance. After the coloured pigment separation solution was transferred into the volumetric flask. Spectrophotometer was used to record the absorbance at 452 nm, and a blank was prepared using petroleum ether. Against standard curve reading was plotted and displayed as 'mg 100 $g^{-1}$ FW'.

### Ascorbic acid (AA) and antioxidant (AOX) activity

AA was determined according to the procedure followed by *Singh et al. (2022)*. It was examined using a 2,6-Dichlorophenol indophenol titration method. The antioxidant (AOX) activity of mango genotypes was estimated using DPPH (2,2-Diphenylpicrylhydrazyl) method followed by *Lu et al. (2014)*. After thoroughly mixing 0.1 mL extract with 3.9 mL of a 0.06 mM DPPH solution mixture was left for thirty minutes in the dark and absorbed the absorbance at 517 nm. The Trolox standard solution was prepared by adding 1 mL of ethanol to the Trolox Standard tube, followed by dissolution by vortexing. The solution was then transferred to a 10 mL measuring flask and ethanol was added to make the final volume 10 mL. The 100 µg/mL solution was then diluted with ethanol to make 80, 60, 40 and 0 µg/mL solutions (*Prasad et al., 2022a*). The AOX activity was expressed as 'µmol Trolox equivalent/g'.

### Total phenolic content (TPC)

Methodology with some modification followed by *Prasad et al. (2022b)* was used to estimate the TPC. Double-distilled water (2.5 mL) in a test tube was used to dilute the (0.5 mL) pulp and then incubated for 3 min after adding 0.5 mL Folin-Ciocalteu reagent. Following incubation, 2 mL of 20% (w/v), $Na_2CO_3$ was added to the sample tube and kept for 1 min for boiling in a water bath. At 650 nm, absorbance was recorded while gallic acid anhydrous standard solutions at concentrations ranging from 10 to 100 mg/L were used to construct a five-point analytical curve. The curves demonstrated satisfactory linearity within the absorbance at each concentration ($R^2 = 0.999$). TPC was displayed as 'µg GAE/g FW.'

*Total flavonoid contents (TFC)*

The TFC was determined per the methodology *Zhisen, Mengcheng & Jianming (1999)* described using aluminium chloride. An extract aliquot of 0.1 ml was taken in 10 ml of a volumetric flask containing 4 ml of distilled water, 0.3 ml of 5% $NaNO_2$ and 0.3 ml of 10% $AlCl_3.6H_2O$. At room temperature mixture was left to stand for 6 min. After adding 2 ml of 1M NaOH, the solution was diluted up to 10 ml using distilled water. The mixture was mixed using a vortex. The absorbance was recorded immediately at 510 nm in a spectrophotometer (Model: IG 94UV; IGENE LABSERVE, New Delhi, India). Catechin hydrate standard curve at concentrations between 50 and 300 mg/L was used to calculate the calibration curve ($R^2$ = 0.999). TFC was expressed as 'µg/100 g FW'.

## Determination of minerals

The minerals were determined according to the methodology followed by *Drozdz, Seziene & Pyrzynska (2018)*. Mango fruit samples were digested with di acids (nitric acid and perchloric acid) in ultrapure water (Milli Q system Millipore, Molsheim, France) to estimate the mineral content. The Milli Q system was utilised for further dilution and digested the samples. The phosphorous reading was recorded in a Spectrophotometer (IGENE LABSERVE, New Delhi, India). A Flame photometer (model no. SP-V325) was used for the minerals such as calcium and potassium and was expressed as 'mg/kg'.

## Physiological attributes

The respiration rate was measured with an automated gas analyser (Model PBI Densor), as described by *Prasad et al. (2022b)*. High-precision electronic balance was used for the determination of the PLW of fruits. Fruits that exhibited more than 10% PLW loss were deemed to have a shelf life completed (*Prasad et al., 2022a*).

## Organoleptic evaluation

Mango genotypes were evaluated for the organoleptic parameter using the 'panel method' and hedonic scale (*Prasad, Jacob & Siddiqui, 2018*). Mango fruit with superior flavour, texture and colour displayed high-level consumer liking (*Prasad et al., 2022a*).

## Determination of fruit softening enzymes

Polygalactouronase (PG), pectin methylesterase (PME) and lipoxygenase (LOX) activity were determined according to the procedure followed by *Prasad et al. (2022a)* with slight modification.

## Statistical analysis

The investigation was conducted in CRD (completely randomised design) with three replicates. Using one-way analysis of variance analysis of all parameters data was done between different mango genotypes using SAS software. Results comparison was made by calculating critical difference and DMRT (Duncan's multiple range test) at a 5% significance level. In the column, the data were expressed as mean ± standard deviation.
## RESULTS AND DISCUSSION

### Physical attributes

The physical characteristics of horticultural produce play a significant role in designing the system for grading, transport, processing, and packaging (*Prasad et al., 2022b*). Fruit appearance, influenced by colour, size, and shape, is one of the consumers' first and most important factors when purchasing (*Kumar et al., 2018*). Therefore, various physical characteristics were recorded in different mango genotypes. Our finding displayed that there were significant variations in the weight of the mango. The highest fruit weight was observed in 'Fazli' (404.67 g) while the lowest was in 'Gulabkhas' (121.57 g). The peel thickness of mango genotypes was also significant, the highest in 'Fazli' (1.85 mm) and the lowest in 'Chausa' (0.62 mm). Similar differences were observed in the seed weight value, which was found to be the highest in 'Fazli' (27.23 g) while the lowest was in 'Dashehari' (11.97 g). One of the most crucial quality traits of any fruit which determines consumer appeal is fruit firmness and it was observed the highest in 'Mahmood Bahar' (9.37 N) while the lowest was observed in 'Himsagar' (4.13 N). Different fruit shape was observed in different selected mango genotypes (Table 1). The genotypes with higher peel thickness and firmness exhibited higher shelf life. The variations in fruit weight, peel thickness, fruit firmness, seed weight and fruit morphology might be due to genetic differences of genotypes. *Bora, Singh & Singh (2017)* and *Totad et al. (2020)* reported similar variations in the physical characteristics of different mango genotypes served as the basis of this investigation. The difference in physical features is also investigated by *Gentile et al. (2019)*.

### Biochemical attributes

#### Soluble solid content (SSC), total sugars and titratable acidity (TA)

In addition to being the primary ingredients in sweet and sour flavours, SSC and TA are also essential indicators of fruit maturity and postharvest fruit flavour assessment during storage (*Zhao et al., 2021*). SSC in fruits is a crucial quality characteristic linked to composition and texture (*Hossain et al., 2014*; *Prasad et al., 2022a*). Our findings displayed that the SSC was reported the highest in 'Amrapali' (25.20 °B), whereas the lowest SSC was reported in Totapuri (16.20 °B). The total sugars were observed maximum in 'Mallika' (20.12%), while the lowest was observed in Zardalu (14.37%). Fruits' overall taste is related to titratable acidity. Our findings revealed that the titratable acidity was found the highest in 'Malda' (0.37%) while the lowest was found in 'Himsagar' (0.12%) (Table 2A). It has been reported that the relationship between SSC and TA is critical for determining the consumer acceptability of many fruits. Our results were per *Samal et al. (2012)* and *Singh et al. (2022)*, who have reported considerable differences in titratable acidity, soluble solid contents and total sugars among mango genotypes.

#### Ascorbic acid (AA), antioxidant (AOX) activity and total carotenoid

AA is a crucial quality characteristic of fruits and is particularly valued for its antioxidant properties & AOX protects against the occurrence of oxidative stress (*Prasad, Sharma & Srivastav, 2016*; *Prasad et al., 2022b*). The ascorbic acid was observed the highest in 'Jawahar' (44.20 mg/100 g pulp), whereas the lowest was observed in 'Ratna' (14.50 mg/

**Table 1 Variation in physical attributes and organoleptic score of different mango genotypes at peak ripening stage.**

**Attributes**

| Genotypes | Fruit firmness (N) | Peel thickness (mm) | Fruit weight (g) | Seed weight (g) | Fruit shape | Organoleptic evaluation (1−9) |
|---|---|---|---|---|---|---|
| Alphonso | 5.11 ± 0.18[gh] | 0.64 ± 0.02[jk] | 155.80 ± 22.11[fg] | 14.50 ± 1.67[ef] | Oval | 8.00 ± 0.28[cdef] |
| Amrapali | 4.98 ± 0.13[ghi] | 0.73 ± 0.02[i] | 163.03 ± 24.07[fg] | 18.67 ± 1.56[bcd] | Ovate oblong | 9.00 ± 0.24[a] |
| Bombai | 4.27 ± 0.17[kl] | 1.14 ± 0.05[ef] | 149.77 ± 15.43[g] | 14.33 ± 1.11[ef] | Oblong | 7.80 ± 0.31[ef] |
| Chausa | 4.83 ± 0.13[ghij] | 0.62 ± 0.02[k] | 259.20 ± 27.72[c] | 25.70 ± 3.41[a] | Ovate-oval oblique | 8.30 ± 0.22[abcdef] |
| Dashehari | 5.27 ± 0.21[g] | 1.09 ± 0.04[fg] | 138.10 ± 18.83[g] | 11.97 ± 0.75[f] | Oblong | 8.20 ± 0.33[bcdef] |
| Fazli | 5.72 ± 0.23[f] | 1.85 ± 0.07[a] | 404.67 ± 6.96[a] | 27.23 ± 4.51[a] | Long-oval | 8.80 ± 0.35[ab] |
| Gulabkhas | 4.81 ± 0.25[ghij] | 1.21 ± 0.06[e] | 121.57 ± 13.95[g] | 12.57 ± 1.70[f] | Round to oval | 8.50 ± 0.44[abcde] |
| Himsagar | 4.13 ± 0.21[l] | 0.74 ± 0.04[i] | 155.23 ± 17.98[fg] | 17.83 ± 3.23[bcde] | Ovate | 7.90 ± 0.41[def] |
| Jawahar | 6.14 ± 0.49[ef] | 1.04 ± 0.08[g] | 195.37 ± 6.87[ef] | 16.10 ± 1.15[cdef] | Oblong | 8.60 ± 0.68[abcd] |
| Krishna Bhog | 6.87 ± 0.36[d] | 1.10 ± 0.06[fg] | 266.20 ± 21.42[c] | 16.70 ± 1.87[cde] | Round | 8.30 ± 0.43[abcdef] |
| Langra | 6.44 ± 0.13[de] | 0.75 ± 0.02[i] | 215.43 ± 7.63[de] | 18.23 ± 0.90[bcde] | Oval | 7.90 ± 0.16[def] |
| Mahmood Bahar | 9.37 ± 0.37[a] | 1.51 ± 0.06[c] | 164.13 ± 19.09[fg] | 16.57 ± 0.51[cde] | Obliquely-oval | 8.70 ± 0.35[abc] |
| Malda | 4.36 ± 0.09[jkl] | 0.73 ± 0.01[i] | 244.00 ± 21.04[cd] | 19.03 ± 1.22[bcd] | Round | 9.00 ± 0.18[a] |
| Mallika | 7.37 ± 0.58[c] | 0.64 ± 0.05[jk] | 346.00 ± 31.00[b] | 21.30 ± 0.70[b] | Ovate-oblong | 8.80 ± 0.70[ab] |
| Prabha Sankar | 6.76 ± 0.23[d] | 1.41 ± 0.05[d] | 331.73 ± 55.41[b] | 21.67 ± 0.51[b] | Ovate-oblong | 7.70 ± 0.27[f] |
| Ratna | 4.35 ± 0.09[jkl] | 0.89 ± 0.02[h] | 255.00 ±2 0.58[cd] | 17.00 ± 2.35[cde] | Ovate oblong to oval | 8.80 ± 0.18[ab] |
| Safed Malda | 4.54 ± 0.12[ijkl] | 0.71 ± 0.02[ij] | 244.00 ± 22.03[cd] | 19.00 ± 1.35[bcd] | Round | 8.40 ± 0.22[abcdef] |
| Suvarnarekha | 8.51 ± 0.34[b] | 1.04 ± 0.04[g] | 266.37 ± 28.55[c] | 18.90 ± 2.26[bcd] | Ovate-oblong | 8.60 ± 0.34[abcd] |
| Totapuri | 4.71 ± 0.19[hijk] | 1.61 ± 0.06[b] | 215.43 ± 4.22[de] | 20.10 ± 3.90[bc] | Oblong | 8.40 ± 0.34[abcdef] |
| Zardalu | 8.42 ± 0.22[b] | 1.11 ± 0.03[fg] | 243.57 ± 20.91[cd] | 15.17 ± 0.49[def] | Oblong-obliquely oblong | 8.10 ± 0.21[bcdef] |
| LSD at 5% | 0.45 | 0.08 | 38.09 | 14.50 | | 0.60 |

**Note:**
Results are means of three determinations ± standard deviations. Mean values in a column with the same alphabetic letters are not significantly different as per Duncan's Multiple Range test.

100 g pulp) (Table 2B). Significant differences in AOX activity existed between the studied mango genotypes, and it was found to be the highest in 'Malda' (5.79 µmol TE/g) while the lowest in 'Totapuri' (2.54 µmol TE/g) (Table 2B). Carotenoids in mango contributed to antioxidant properties. The mango genotypes varied in total carotenoid contents and it was reported the highest TCC in 'Amrapali' (9.10 mg/100 g) and the lowest in 'Langra' (5.50 mg/100 g) (Table 2B). The genotypes rich in these biochemical compounds are highly preferred by consumers. The higher antioxidant value of genotypes might be due to higher levels of total phenol, ascorbic acid and total carotenoid content. This study got

**Table 2A Variation in biochemical attributes of different mango genotypes at peak ripening stage.**

**Attributes**

| Genotypes | Soluble solids contents (%) | Total sugars (%) | Titratable acidity (%) |
|---|---|---|---|
| Alphonso | 21.80 ± 0.76[c] | 18.14 ± 0.63[bcd] | 0.25 ± 0.009[abcd] |
| Amrapali | 25.20 ± 0.67[a] | 19.37 ± 0.51[ab] | 0.14 ± 0.004[cd] |
| Bombai | 16.30 ± 0.65[i] | 15.41 ± 0.62[ghij] | 0.21 ± 0.008[abcd] |
| Chausa | 22.00 ± 0.58[c] | 15.31 ± 0.41[hij] | 0.21 ± 0.006[abcd] |
| Dashehari | 22.80 ± 0.91[bc] | 17.02 ± 0.68[def] | 0.29 ± 0.012[abcd] |
| Fazli | 17.00 ± 0.68[hi] | 16.05 ± 0.64[fghi] | 0.32 ± 0.007[abcd] |
| Gulabkhas | 19.20 ± 1.00[efg] | 16.04 ± 0.83[fghi] | 0.22 ± 0.011[abcd] |
| Himsagar | 20.30 ± 1.05[de] | 16.57 ± 0.86[efgh] | 0.12 ± 0.006[d] |
| Jawahar | 18.00 ± 1.43[fgh] | 17.11 ± 1.36[def] | 0.24 ± 0.019[abcd] |
| Krishna Bhog | 17.80 ± 0.92[ghi] | 18.82 ± 0.98[bc] | 0.16 ± 0.008[bcd] |
| Langra | 21.60 ± 0.43[cd] | 16.75 ± 0.34[efg] | 0.21 ± 0.004[abcd] |
| Mahmood Bahar | 19.40 ± 0.78[ef] | 15.06 ± 0.60[ij] | 0.34 ± 0.013[ab] |
| Malda | 18.00 ± 0.36[fgh] | 17.20 ± 0.34d[ef] | 0.37 ± 0.013[a] |
| Mallika | 23.70 ± 1.88[b] | 20.12 ± 1.60[a] | 0.17 ± 0.013[bcd] |
| Prabha Sankar | 17.70 ± 0.61[ghi] | 17.61 ± 0.61[cde] | 0.33 ± 0.013[abc] |
| Ratna | 20.10 ± 0.40[e] | 16.21 ± 0.32[efghi] | 0.23 ± 0.005[abcd] |
| Safed Malda | 17.50 ± 0.46[hi] | 15.99 ± 0.42[fghi] | 0.32 ± 0.008[abcd] |
| Suvarnarekha | 17.60 ± 0.70[hi] | 15.21 ± 0.61[hij] | 0.35 ± 0.014[ab] |
| Totapuri | 16.20 ± 0.52[i] | 15.34 ± 0.61[ghij] | 0.19 ± 0.008[abcd] |
| Zardalu | 22.50 ± 0.23[bc] | 14.37 ± 0.38[j] | 0.17 ± 0.004[bcd] |
| LSD at 5% | 1.41 | 1.23 | 0.02 |

**Note:**
Results are means of three determinations ± standard deviations. Mean values in a column with the same alphabetic letters are not significantly different as per Duncan's Multiple Range test.

evidence from the results of *Gentile et al. (2019)* and *Lu et al. (2014)*, who reported variations in biochemical attributes in selected mango cultivars.

## Total phenolic and total flavonoid contents

Dietary antioxidants-derived phenolics lower the risk of various chronic diseases (*Jayarajan et al., 2019*). Flavour (different shades of flavour) and antioxidant activity of fruits are attributed to flavonoids. The concentration of phenolics is variable among the different mango genotypes, being the highest in 'Malda' (560.60 μg/100 g) while the lowest was registered in 'Totapuri' (297.50 μg/100 g). The highest TFC were observed in Mallika (700.00 μg/g), whereas the lowest TFC was observed in 'Himsagar' (355.00 μg/g) (Table 2B). The genotypes with a high TPC and TFC are considered to have high fruit quality and nutraceutical value (*Rastegar, Hassanzadeh Khankahdani & Rahimzadeh, 2019*). Such differences between TCC and TFC also have been reported by *Totad et al. (2020)* in blueberry varieties grown in the northern-western Himalayas.

**Table 2B Biochemical attributes of different mango genotypes at peak ripening stage.**

**Attributes**

| Genotypes | Total phenolic content (µg/100 g) | Total flavonoid (µg/g) | Ascorbic acid (mg/100 g pulp) | Antioxidant acidity (µmol TE/g) | Total carotenoid (mg/100 g) |
|---|---|---|---|---|---|
| Alphonso | 475.50 ± 16.47[cd] | 407.00 ± 14.10[ij] | 20.40 ± 0.71[i] | 3.87 ± 0.13[a] | 6.60 ± 0.23[e] |
| Amrapali | 443.30 ± 11.73[def] | 590.00 ± 15.61[de] | 24.40 ± 0.65[gh] | 5.42 ± 0.14[b] | 9.10 ± 0.24[a] |
| Bombai | 469.30 ± 18.77[cd] | 367.00 ± 14.68[jk] | 20.80 ± 0.83[i] | 3.74 ± 0.15[e] | 5.60 ± 0.22[g] |
| Chausa | 300.10 ± 7.94[j] | 530.00 ± 14.02[fg] | 15.30 ± 0.40[j] | 3.25 ± 0.09[h] | 5.60 ± 0.15[g] |
| Dashehari | 417.50 ± 16.70[fg] | 387.40 ± 15.50[jk] | 28.60 ± 1.14[ef] | 4.63 ± 0.19[c] | 8.40 ± 0.34[b] |
| Fazli | 510.60 ± 20.42[b] | 404.00 ± 16.16[ij] | 30.30 ± 1.21[e] | 3.41 ± 0.14[fgh] | 8.20 ± 0.33[bc] |
| Gulabkhas | 403.50 ± 20.97[gh] | 431.00 ± 22.40[hi] | 29.90 ± 1.55[e] | 3.84 ± 0.20[e] | 6.20 ± 0.32[ef] |
| Himsagar | 343.10 ± 17.83[i] | 355.00 ± 18.45[k] | 15.30 ± 0.80[j] | 3.73 ± 0.19[ef] | 7.30 ± 0.38[d] |
| Jawahar | 384.60 ± 30.53[h] | 512.00 ± 40.64[g] | 44.20 ± 3.51[a] | 3.33 ± 0.26[gh] | 6.70 ± 0.53[e] |
| Krishna Bhog | 452.20 ± 23.50[de] | 450.00 ± 23.38[h] | 23.30 ± 1.21[h] | 3.56 ± 0.18[efgh] | 5.80 ± 0.30[fg] |
| Langra | 353.00 ± 7.06[i] | 550.30 ± 11.01[fg] | 41.40 ± 0.83[b] | 3.65 ± 0.07[efg] | 5.50 ± 0.11[g] |
| Mahmood Bahar | 333.00 ± 13.32[ij] | 563.00 ± 22.52[ef] | 26.60 ± 1.06[fg] | 4.17 ± 0.17[d] | 8.70 ± 0.35[ab] |
| Malda | 560.60 ± 11.21[a] | 620.00 ± 12.40[cd] | 40.40 ± 0.81[bc] | 5.79 ± 0.02[a] | 8.60 ± 0.17[ab] |
| Mallika | 421.30 ± 33.44[efg] | 700.00 ± 55.56[a] | 28.60 ± 2.27[ef] | 5.24 ± 0.42[b] | 8.60 ± 0.68[ab] |
| Prabha Sankar | 456.60 ± 15.82[d] | 450.00 ± 15.59[h] | 28.30 ± 0.98[ef] | 5.11± 0.18[b] | 6.30 ± 0.22[ef] |
| Ratna | 513.30 ± 10.27[b] | 611.00 ± 12.22[d] | 14.50 ± 0.29[j] | 3.87 ± 0.08[de] | 8.40 ± 0.17[b] |
| Safed Malda | 500.00 ± 13.23[bc] | 657.00 ± 17.38[bc] | 37.50 ± 0.99[d] | 4.51 ± 0.12[c] | 8.50 ± 0.22[b] |
| Suvarnarekha | 467.80 ± 18.71[d] | 660.00 ± 26.40[b] | 39.70 ± 1.59[bcd] | 4.75 ± 0.19[c] | 7.70 ± 0.31[cd] |
| Totapuri | 297.50 ± 11.90[j] | 467.00 ± 18.68[h] | 28.60 ± 1.14[ef] | 2.54 ± 0.10[i] | 6.30 ± 0.25[ef] |
| Zardalu | 412.20 ± 10.91[fgh] | 540.00 ± 14.29[fg] | 38.40 ± 1.02[cd] | 4.19 ± 0.11[d] | 8.80 ± 0.23[ab] |
| LSD at 5% | 29.57 | 37.42 | 2.22 | 0.30 | 0.52 |

**Note:**
Results are means of three determinations ± standard deviations. Mean values in a column with the same alphabetic letters are not significantly different as per Duncan's Multiple Range test.

## Physiological attributes

The attributes such as respiration rate (ml $CO_2$ kg$^{-1}$ h$^{-1}$), physiological loss in weight (PLW) (%) and shelf life (days) differed significantly among selected mango genotypes. Respiration rate is a crucial factor affecting a fruit's shelf life (*Jhalegar, Sharma & Singh, 2014*). The maximum respiration rate among mango genotypes was reported in 'Chausa' (137.19 ml $CO_2$ kg$^{-1}$ h$^{-1}$), while the lowest was observed in 'Safed Malda' (95.44 77 ml $CO_2$ kg$^{-1}$ h$^{-1}$). PLW is assessed by moisture loss from the fruit due to transpiration or respiration, which is governed by fruit peel thickness or environmental factors. The highest PLW was observed in 'Totapuri' (13.00), whereas the lowest PLW was in 'Amrapali' (6.20%). The highest shelf life was observed in 'Malda', 'Safed Malda' and 'Suvarnarekha' (11–12 days) (Table 3). The genotypes exhibited higher respiration rates and PLW had a lower shelf life. A similar study investigated kiwi genotypes (*Sharma et al., 2015*) and mango (*Prasad et al., 2022a*).

**Table 3 Differences in physiological attributes of different mango genotypes at peak ripening stage.**

| Attributes | | | |
|---|---|---|---|
| Genotypes | Respiration rate (ml $CO_2$ $kg^{-1}$ $h^{-1}$) | Physiological in weight (%) | Peak ripening stage (on day) | Shelf life (days) |
|---|---|---|---|---|
| Alphonso | $107.27 \pm 3.72^{cdef}$ | $6.40 \pm 0.22^{i}$ | 06 | 08–10 |
| Amrapali | $101.09 \pm 2.67^{fgh}$ | $6.20 \pm 0.16^{i}$ | 08 | 10–12 |
| Bombai | $110.26 \pm 4.41^{cdef}$ | $11.90 \pm 0.48^{b}$ | 05 | 06–08 |
| Chausa | $137.19 \pm 3.63^{a}$ | $10.00 \pm 0.26^{cd}$ | 05 | 06–08 |
| Dashehari | $111.37 \pm 4.45^{cde}$ | $8.40 \pm 0.34^{fgh}$ | 06 | 08–10 |
| Fazli | $102.12 \pm 4.08^{efgh}$ | $8.60 \pm 0.34^{fg}$ | 06 | 08–10 |
| Gulabkhas | $121.32 \pm 6.30^{b}$ | $10.60 \pm 0.55^{c}$ | 05 | 06–08 |
| Himsagar | $134.81 \pm 7.00^{a}$ | $8.30 \pm 0.43^{gh}$ | 05 | 07–09 |
| Jawahar | $105.61 \pm 8.38^{defg}$ | $9.50 \pm 0.75^{de}$ | 05 | 07–09 |
| Krishna Bhog | $112.43 \pm 5.84^{bcd}$ | $8.50 \pm 0.44^{fgh}$ | 07 | 09–11 |
| Langra | $115.63 \pm 2.31^{bc}$ | $6.30 \pm 0.13^{i}$ | 07 | 09–11 |
| Mahmood Bahar | $110.55 \pm 4.42^{cdef}$ | $8.80 \pm 0.35^{efg}$ | 08 | 10–12 |
| Malda | $97.54 \pm 1.95^{gh}$ | $6.60 \pm 0.13^{i}$ | 08 | 10–12 |
| Mallika | $131.36 \pm 10.43^{a}$ | $8.50 \pm 0.67^{fgh}$ | 06 | 08–10 |
| Prabha Sankar | $120.61 \pm 4.18^{b}$ | $9.10 \pm 0.32^{ef}$ | 06 | 08–10 |
| Ratna | $104.50 \pm 2.09^{defgh}$ | $7.80 \pm 0.16^{h}$ | 06 | 08–10 |
| Safed Malda | $95.44 \pm 2.53^{h}$ | $9.50 \pm 0.25^{de}$ | 08 | 10–12 |
| Suvarnarekha | $96.65 \pm 3.87^{gh}$ | $9.10 \pm 0.36^{ef}$ | 08 | 10–12 |
| Totapuri | $135.72 \pm 5.43^{a}$ | $13.00 \pm 0.52^{a}$ | 05 | 05–07 |
| Zardalu | $109.77 \pm 2.90^{cdef}$ | $8.70 \pm 0.23^{fg}$ | 05 | 07–09 |
| LSD at 5% | 8.30 | 0.65 | | |

**Note:**
Results are means of three determinations ± standard deviations. Mean values in a column with the same alphabetic letters are not significantly different as per Duncan's Multiple Range test.

## Organoleptic evaluation

Significant differences were observed for organoleptic evaluation among selected genotypes of mango. The highest organoleptic score was reported in 'Amrapali' (9.0), while the lowest was observed in 'Prabha Sankar' (7.0). The greater sensory score in mango genotypes might be due to improved colour, taste, fragrance and flavour (*Prasad & Sharma, 2018*; *Prasad et al., 2022b*) (Table 2A).

## Minerals

Minerals are necessary for the body's healthy operation, growth and development, and preserving health. Potassium is related to fruit quality, phosphorous stabiles fruit cell walls and calcium is needed to keep cells rigid (*Sinha et al., 2017*). Considerable variations in primary mineral constituents among different genotypes of mango were observed. Among the genotypes evaluated, potassium content was reported the highest in 'Bombai' (12.46 mg/kg), whereas the lowest was observed in 'Mahmood Bahar' (5.60 mg/kg). Apart

**Table 4 Variation in mineral content of different mango genotypes at peak ripening stage.**

**Attributes**

| Genotypes | Potassium (mg/kg) | Phosphorous (mg/kg) | Calcium (mg/kg) |
|---|---|---|---|
| Alphonso | 9.80 ± 0.34[ef] | 1.73 ± 0.06[abcd] | 0.18 ± 0.004[bcde] |
| Amrapali | 9.70 ± 0.26[ef] | 1.75 ± 0.05[abc] | 0.15 ± 0.004[cde] |
| Bombai | 12.46 ± 0.50[a] | 1.63 ± 0.07[bdef] | 0.13 ± 0.005[e] |
| Chausa | 11.30 ± 0.30[bc] | 1.61 ± 0.04[def] | 0.16 ± 0.006cde |
| Dashehari | 9.70 ± 0.26[ef] | 1.67 ± 0.07[abcde] | 0.18 ± 0.014[bcde] |
| Fazli | 9.90 ± 0.40[ef] | 1.62 ± 0.06[def] | 0.20 ± 0.004[bcde] |
| Gulabkhas | 10.30 ± 0.41[de] | 1.05 ± 0.05[h] | 0.14 ± 0.004[de] |
| Himsagar | 10.70 ± 0.56[cd] | 1.04 ± 0.05[h] | 0.11 ± 0.004[e] |
| Jawahar | 8.70 ± 0.23[g] | 1.57 ± 0.12[ef] | 0.24 ± 0.012[abcde] |
| Krishna Bhog | 9.40 ± 0.75[fg] | 1.63 ± 0.08[bdef] | 0.29 ± 0.012[abcde] |
| Langra | 9.60 ± 0.50[ef] | 0.93 ± 0.02[h] | 0.22 ± 0.011[abcde] |
| Mahmood Bahar | 5.60 ± 0.11[i] | 1.65 ± 0.07[abcde] | 0.40 ± 0.014[a] |
| Malda | 6.60 ± 0.13[h] | 1.77 ± 0.04[a] | 0.12 ± 0.010[e] |
| Mallika | 8.70 ± 0.35[g] | 1.52 ± 0.12[f] | 0.33 ± 0.007[abcd] |
| Prabha Sankar | 12.30 ± 0.33[a] | 1.59 ± 0.06[ef] | 0.26 ± 0.007[abcde] |
| Ratna | 11.20 ± 0.45[bc] | 1.60 ± 0.03[ef] | 0.11 ± 0.006[e] |
| Safed Malda | 11.80 ± 0.24[ab] | 1.75 ± 0.05[ab] | 0.13 ± 0.005[e] |
| Suvarnarekha | 11.50 ± 0.23[b] | 1.76 ± 0.07[a] | 0.36 ± 0.029[ab] |
| Totapuri | 11.10 ± 0.88[bc] | 1.35 ± 0.05[g] | 0.14 ± 0.003[de] |
| Zardalu | 9.20 ± 0.37[fg] | 1.67 ± 0.04[abcde] | 0.34 ± 0.009[abc] |
| LSD at 5% | 0.70 | 0.11 | 0.02 |

**Note:**
Results are means of three determinations ± standard deviations. Mean values in a column with the same alphabetic letters are not significantly different as per Duncan's Multiple Range test.

from potassium, phosphorous contents also showed considerable variation among different mango genotypes, which was observed the highest in 'Malda' (1.77 mg/kg). In comparison, the lowest was observed in Langra (0.93 mg/kg). The value of calcium among selected mango genotypes was registered the highest in 'Mahmood Bahar' (0.40 mg/kg) while the lowest was registered in 'Ratna' and 'Himsagar' (0.11 mg/kg) (Table 4). The genotypes with higher mineral contents are considered nutritionally rich (*Kumar et al., 2018*). A similar finding has been investigated by *Akin-Idowu et al. (2020)* in different fruit and *Lu et al. (2014)* in pineapple cultivars. *Drozdz, Seziene & Pyrzynska (2018)* and *Totad et al. (2020)* also have reported such differences in minerals among selected wild and cultivated blueberry genotypes, respectively.

## Fruit softening enzymes

Fruit softening enzymes such as PG enzyme (μg galacturonic acid $g^{-1}$ $h^{-1}$ FW), PME (μmol $g^{-1}$ FW $min^{-1}$) and LOX (μmol $g^{-1}$ FW $min^{-1}$) varied significantly among selected mango genotypes. PG and PME enzymes are directly associated with fruit ripening, softening and textural changes processes and cell wall decomposition, while LOX is

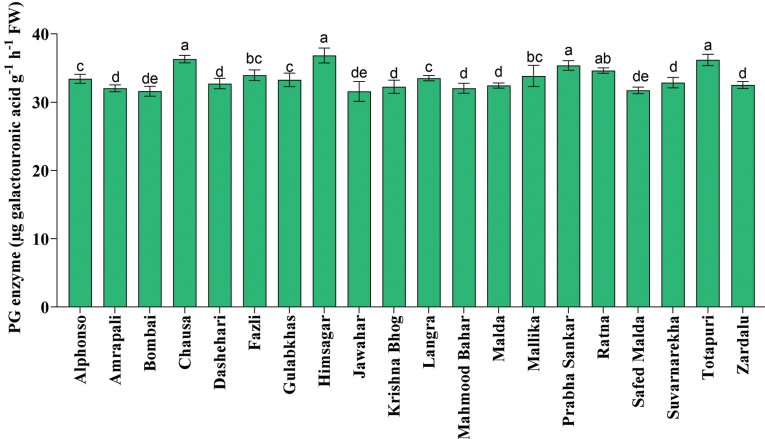

**Figure 1 Polygalactouronase enzymatic (PG) activity.** Variation in polygalactouronase enzymatic (PG) activity of different mango genotypes at peak ripening stage. Lowercase letters indicate significant differences between treatments ($p < 0.05$).

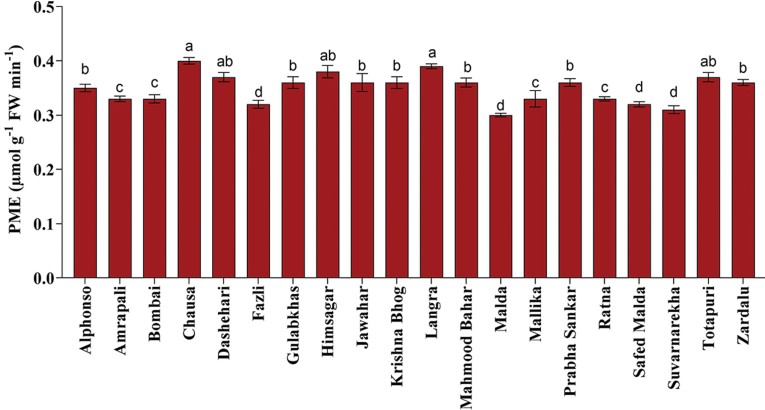

**Figure 2 Pectin methylesterase enzymatic activity.** Variation in pectin methylesterase enzymatic activity of different mango genotypes at peak ripening stage. Lowercase letters indicate significant differences between treatments ($p < 0.05$).

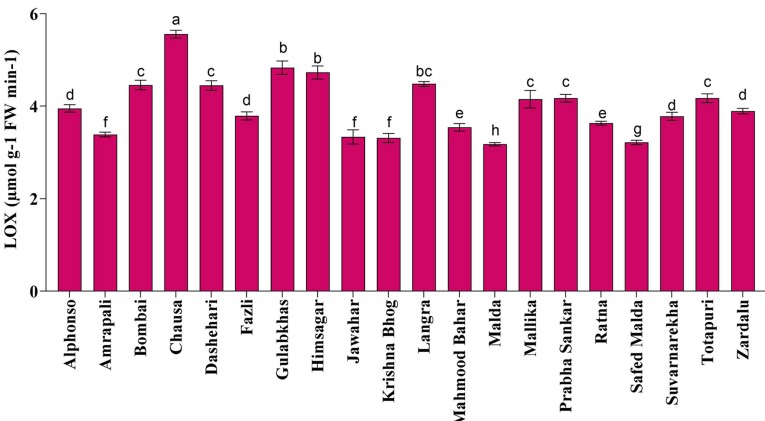

**Figure 3 Lipoxygenase enzymatic activity.** Differences in lipoxygenase enzymatic activity of different mango genotypes at peak ripening stage. Lowercase letters indicate significant differences between treatments ($p < 0.05$).

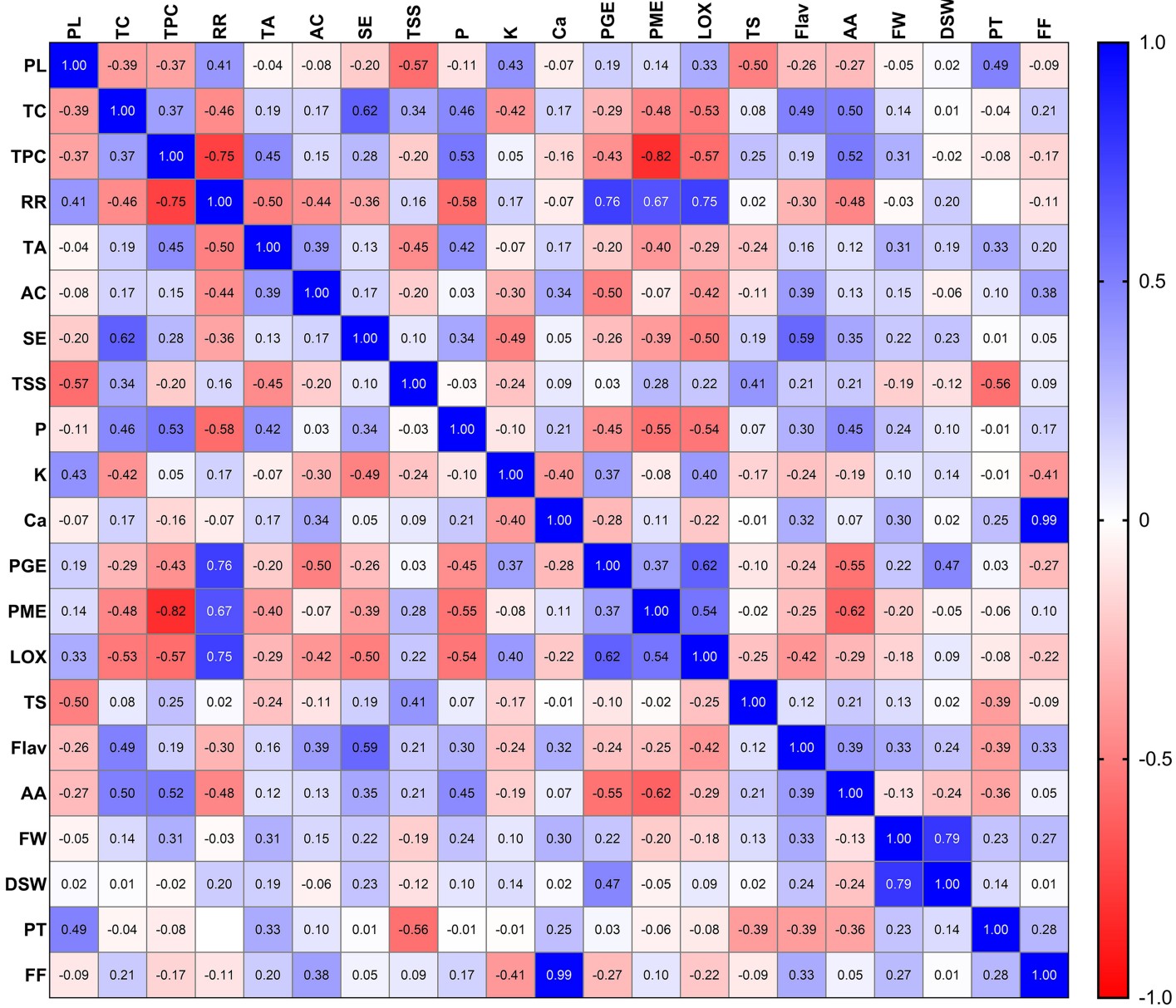

**Figure 4 Pearson correlation matrix.** Pearson correlation matrix between all the twenty-one parameters of mango quality traits. Abbreviations: PL, physiological loss in weight; TC, total carotenoid; TPC, total phenol content: RR, respiration rate; TA, titrable acidity; AC, ascorbic acid; SE, sensory evaluation; TSS, total soluble solid; P, phosphorus; K, potassium; Ca, calcium; PGE, polygalacturonase; PME, pectin methylesterase; LOX, lipoxygenase; TS, total sugar; Flav, flavanoid content; AA, antioxidant activity; FW, fruit weight; DSW, dry seed weight; PT, peel thickness; FF, fruit firmness.

associated with senescence. The PG enzyme activity was found the highest in 'Himsagar' (36.83 μg galacturonic acid g$^{-1}$ h$^{-1}$ FW), whereas, the lowest was found in 'Jawahar' (31.57 μg galacturonic acid g$^{-1}$ h$^{-1}$ FW) (Fig. 1). The PME activity was observed maximum in 'Chausa' (0.40 μmol g$^{-1}$ FW min$^{-1}$) while the lowest in 'Malda' (0.30 μmol g$^{-1}$ FW min$^{-1}$) (Fig. 2). The LOX activity was registered the highest in 'Chausa' (5.56 μmol g$^{-1}$ FW min$^{-1}$). In contrast, the lowest was observed in 'Malda' (3.18 μmol g$^{-1}$ FW min$^{-1}$) (Fig. 3).

This study got the support of *Prasad et al. (2020)* who observed considerable variations for fruit softening enzymes among genotypes of mango.

## Correlation analysis of parameter for quality traits in mango

The correlation analysis of the selected twenty-one different traits in mango genotypes at their peak ripening stage revealed an overall significant positive correlation (one variable increases with the increase in another variable) among the physical attributes of different mango genotypes (Fig. 4). The soluble solids contents, total sugars, total carotenoid and organoleptic evaluation also showed positive correlations with each other when compared across different mango genotypes at peak ripening stage. Similarly, total phenolic content, total flavonoid, ascorbic acid, and antioxidant acidity were also found to be positively correlated with each other (Fig. 4). Also, the physiological attributes such as respiration rate and physiological loss in weight, were also suggested to be positively correlated with each other. This finding reveals that genotypes exhibiting lower respiration rates and physiological loss in weight exhibit higher shelf life. The total phenol content was negatively correlated with PME, PG and LOX activity. In contrast, the total carotenoid content was positively correlated with sensory evaluation (Fig. 4). This finding suggests that better colour development in some genotypes due to carotenoid content is responsible for the higher acceptability of that genotype by consumers. The valuable insights into the correlations between various traits of mango genotypes at the peak ripening stage may aid in selecting and breeding superior mango cultivars with desirable traits and characteristics. Overall, this study provides valuable insights into the correlations between various traits of mango genotypes at the peak ripening stage. These findings may aid in selecting and breeding superior mango cultivars with desirable traits and characteristics.

## CONCLUSION

This study showed significant variations the bioactive and fruit quality compounds of the studied mango genotypes. 'Malda' was found to be superior in terms of total phenolic content (560.60 µg/100 g), total antioxidant (5.79 µmol TE/g), and titratable acidity (0.37%). 'Amrapali' had the highest soluble solid content (25.20 °B), 'Jawahar' had the highest ascorbic acid content (44.20 mg/100 g pulp), 'Mallika' had the highest total flavonoid content (700.00 µg/g), and 'Amrapali' had the highest total carotenoid content (9.10 mg/100 g). Genotypes such as 'Malda', 'Safed Malda', and 'Suvarnarekha' exhibited higher shelf life, indicating their potential for use in processing and storage. The genotypes with higher biochemical content are considered to have high nutraceutical value.

The genotypes that exhibited higher TSS and total sugars can be preferred for juice processing. The superior genotypes regarding bioactive and fruit-quality compounds can be recommended for fresh consumption. Additionally, the practical utility of these results extends to the quality improvement program and processing industry, where the findings can be used to improve the quality and value of mango products.

### Funding

The authors received no funding for this work.

### Competing Interests

Dr. Ravinder Kumar is an Academic Editor for PeerJ.

### Author Contributions

- Neetu Saroj conceived and designed the experiments, performed the experiments, analyzed the data, prepared figures and/or tables, authored or reviewed drafts of the article, and approved the final draft.
- K. Prasad conceived and designed the experiments, performed the experiments, analyzed the data, prepared figures and/or tables, authored or reviewed drafts of the article, and approved the final draft.
- Sanjay Kumar Singh conceived and designed the experiments, performed the experiments, analyzed the data, authored or reviewed drafts of the article, and approved the final draft.
- Vishal Kumar conceived and designed the experiments, performed the experiments, analyzed the data, prepared figures and/or tables, and approved the final draft.
- Shubham Maurya performed the experiments, analyzed the data, prepared figures and/or tables, and approved the final draft.
- Poonam Maurya performed the experiments, analyzed the data, prepared figures and/or tables, authored or reviewed drafts of the article, and approved the final draft.
- Rahul Kumar Tiwari conceived and designed the experiments, analyzed the data, prepared figures and/or tables, authored or reviewed drafts of the article, and approved the final draft.
- Milan Kumar Lal conceived and designed the experiments, prepared figures and/or tables, authored or reviewed drafts of the article, and approved the final draft.
- Ravinder Kumar performed the experiments, analyzed the data, authored or reviewed drafts of the article, and approved the final draft.

### Data Availability

Raw data is available in the Supplemental Files.

### Supplemental Information

Supplemental information for this article can be found online at http://dx.doi.org/10.7717/peerj.15867#supplemental-information.

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
