# Peer review of "Characterization of bioactive and fruit quality compounds of promising mango genotypes grown in Himalayan plain region"

_PeerJ, doi:10.7717/peerj.15867_

## Round 0.1 · original submission · Major Revisions

Please provide a revised version and a rebuttal letter. In particular, aside from all important queries, the statistical analysis of the enzymatic activity is fundamental to support conclusions.

Reviewer 1 ·

Basic reporting

Here, the authors make a description of 20 mango genotypes using different parameters such as the physical, physiological, biochemical, mineral and organoleptic attributes. However, despite the data here reported could contribute to expanding knowledge in this area, the work lacks important points that must be considered before being accepted.

Minor comments:

-Lines 51-52: please use italics for scientific names.
-Write abbreviations in parentheses only the first time they are mentioned (i.e. line 207).
Line 207: what do you mean by "differences in AOX"? Do you mean antioxidant capacity or some other parameter? It's not clear.
-Lines 212-213: The sentence is not clear. Do you mean that the antioxidant properties are due to those changes? Please be clear.
-In the conclusions, raw or numerical data is normally no longer included.

Major comments:

-What do you mean by "peak ripening stage"? You need to define this concept because all your results are based on this. In this sense, if you mean the climacteric peak, previously, you should have determined this peak for each of the different genotypes because it cannot be the same for each one. On the other hand, if you refer to the maximum ripening peak, that is, commercial maturity, a specific day must also be determined for each genotype. For instance, for cultivar "Kent" the commercial ripening is at 16 days post-harvest, meanwhile for cv. "Ataulfo" is on day 10 post-harvest.
-Your results lack a discussion, you only make a description of data. It is mandatory to discuss and compare your results for them to be valid.
-What does it mean that there is a positive or negative relationship between the different parameters?

Experimental design

There are some points that the authors should consider:

-It is necessary to specify for how many days the fruits were stored.
-For each genotype, you need to specify at what exactly day the measurements were performed. If you made these measurements on the same day for each genotype, this must also be specified, although they would not be under the same ripening conditions and the data cannot be compared. i.e., the respiration rate is a very changing parameter for each cultivar, thus, we cannot talk about "the maximum respiration rate" if you didn't determinate the climacteric peak.

Validity of the findings

-Your enzyme activity results do not have statistical analysis.

Additional comments

NA

Reviewer 2 ·

Basic reporting

Your abstract needs to clarify which part of the mango (whole fruit, pulp, juice, etc.) physical, physiological, biochemical, mineral, and organoleptic analyzes were made, also point out that organoleptic evaluation was carried out.
Introduction:
Mangiferin is one of the main mango phenols, with 1690.4 mg/kg in peel (Masibo and Qian, 2008), however, there is little information in the introduction, it is recommended to seek more information on this compound, as well as its beneficial effects on health (antioxidant, anticancer, antimicrobial, antiatherosclerotic, antiallergenic, anti-inflammatory, analgesic, antidiabetic, and immunomodulatory among many others).
In general, is clear and unambiguous.

Experimental design

--The authors must report under what environmental conditions the mangoes were harvested (hot, cold, or rainy seasons).

--What was the number of replicates (n), from the chemical analyses?
-- The antioxidant activity should be clearer, they do not specify how the trolox was prepared
-- Line 124 What µmol TE/g mean?
--Line 130 what do several gallic acid (GA) solutions mean? at what concentrations?
-- In Total flavonoid contents (TFC), solutions standard curve was not used?, catechin, quercetin o rutin are generally used.
-- The technique determination of minerals, must have a reference

Validity of the findings

-- Organoleptic evaluation does not belong to the biochemical attributes (table 2), and may belong more to physical attributes (color, taste, fragrance and flavor), the results should be described next to physical attributes.
-- The results of total carotenoids should be changed from Table 2a to Table 2b
-- The Malda genotype, showed the highest antioxidant activity, however, it did not have the highest content of ascorbic acid, carotenoids, or total flavonoids, only total phenols, how do you explain this? There is no correlation, could it be the presence of mangiferin?

Conclusion: The authors do not point out that according to the physical-chemical attributes, which genotype evaluated, they recommend for a quality improvement program and processing industry

References are suitable for the topic developed.
-- Follow references, the year must not be between parentheses:
Drozdz et al.,
Prasad et al.,
Ranganna S.
Saroj N.

Additional comments

Topics as this article are interesting.

---

## Round 0.2 · Minor Revisions

Thanks for addressing the reviewer´s comments. I attached a file with editorial suggestions and some grammar corrections. Please include them at your convenience.

Reviewer 1 ·

Basic reporting

The authors improved the manuscript and attended the comments. I only have some minor revisions:

-In the introduction, you focus only on the mango's bioactive compounds; however, you don’t address the topics related to the fruit quality parameters such as flavor, texture, color, and the importance of the softening enzymes.

-Figure 1-3: Please indicate the statistical analysis that you made and what the literals mean.

Experimental design

NA

Validity of the findings

NA

Additional comments

NA

Reviewer 2 ·

Basic reporting

In abstract there are abbreviations, which are not explained, you should first describe and place the abbreviations in parentheses.

Introduction:
Mangiferin is one of the main mango phenols, with 1690.4 mg/kg in peel (Masibo and Qian, 2008), however, there is little information in the introduction, it is recommended to seek more information on this compound.
In general, is clear and unambiguous.

Experimental design

--What was the number of replicates (n), from the chemical analyses?
-- The antioxidant activity should be clearer, they do not specify how the trolox was prepared

Validity of the findings

Is Ok
Did not make corrections to references
References are suitable for the topic developed.
-- Follow references, the year must not be between parentheses:
Drozdz et al.,
Prasad et al.,
Ranganna S.
Saroj N.

Additional comments

Topics as this article are interesting.

---

## Round 0.3 · accepted · Accept

Thanks for addressing the minor revisions requested. Now your manuscript is accepted in PeerJ.